# ADDRESSING HIGH-DIMENSIONAL CONTINUOUS ACTION SPACE VIA DECOMPOSED DISCRETE POLICY-CRITIC

## ABSTRACT

Reinforcement learning (RL) methods for discrete action spaces like DQNs are being widely used in tasks such as Atari games. However, they encounter difficulties when addressing continuous control tasks, since discretizing continuous action space incurs the curse-of-dimensionality. To tackle continuous control tasks via discretized actions, we propose a decomposed discrete policy-critic (D2PC) architecture, which was inspired by multi-agent RL (MARL) and associates with each action dimension a discrete policy, while leveraging a single critic network to provide a shared evaluation. Building on D2PC, we advocate soft stochastic D2PC (SD2PC) and deterministic D2PC (D3PC) methods with a discrete stochastic or deterministic policy, which show comparable or superior training performances relative to even continuous actor-critic methods. Additionally, we design a mechanism that allows D3PC to interact with continuous actor-critic methods, contributing to the Q-policy-critic (QPC) algorithm, which inherits the training efficiency of discrete RL and the near-optimal final performance of continuous RL algorithms. Substantial experimental results on several continuous benchmark tasks validate our claims.

## 1 INTRODUCTION

Reinforcement learning (RL) Sutton & Barto (2018) is a class of machine learning methods that train models using interactions with the environment. RL can be divided into discrete RL with a discrete action space that contains limited actions, and continuous RL with a continuous action space having infinite actions. Actor-critic Konda & Tsitsiklis (1999), a structure that uses a policy network to learn a state-action value function, is commonly used to address continuous RL tasks such as continuous control, but it is demonstrated to be fragile and sensitive to hyperparameters Haarnoja et al. (2018a).

Employing discrete RL algorithms for continuous control via action discretization is a feasible way to improve the training efficiency Pazis & Lagoudakis (2009), since discrete RL algorithms including Q-learning Watkins & Dayan (1992) and deep Q network (DQN) Mnih et al. (2013; 2015) have low complexity and exhibit stable training behavior. But for continuous control tasks with high-dimensional continuous action spaces Levine et al. (2016), naive action discretization would lead to the dimension explosion problem. For example, if the continuous action space contains $M$ dimensions and each dimension is discretized to $N$ fractions, there will be $M^N$ actions for the discrete RL algorithm.

As a remedy, a possible way to address this problem is to discretize independently each action dimension of the high-dimensional continuous action space, so that the total number of discrete actions $MN$ grows linear instead of exponential in dimension $M$. Nonetheless, this procedure may lead to the low sample-efficiency problem, because one can only take on-policy iterations to collaboratively learn the relationship between different action dimensions' policies. To integrate the discrete action space and off-policy iterations in a continuous RL algorithm, we propose a decomposed discrete policy-critic (D2PC) architecture inspired by multi-agent actor-critic methods, by viewing each action dimension as an independent agent and assigning a uni-dimensional policy which is optimized by referencing a centralized critic network.

Based on D2PC, we propose two algorithms which can effectively address continuous control tasks via discrete policy and experience replay. The first algorithm is soft D2PC (SD2PC), which takes maximum-entropy value iterations and optimizes softmax stochastic policies for each action dimension; and the second called determined D2PC (D3PC), which assigns an independent Q function for each action dimension fitting the critic's value function by supervised learning. Experiments show that the two algorithms exhibit high training efficiencies and markedly outperform state-of-the-art actor-critic algorithms like twin delayed deep deterministic policy gradients (TD3) Fujimoto et al. (2018) or soft actor-critic (SAC) Haarnoja et al. (2018a).

Though algorithms with D2PC trains fast, we observe from experiments that they may be limited in achieving the best performance finally, namely getting stuck in locally optimal solutions. Since D2PC and standard actor-critic methods share the same critic structure, this motivates us to combine the best of the two methods, by developing the Q-policy-critic (QPC) algorithm. Our QPC uses a discrete-continuous hybrid policy, co-trains a continuous actor along with D2PC, and dynamically exploits the discrete and continuous actors to improve the critic network. Substantial numerical tests over continuous control tasks demonstrate that QPC achieves significant improvement in convergence, stability, and rewards compared with both D3PC and continuous actor-critic methods.

## 2 RELATED WORK

As pointed out in Tavakoli et al. (2018), it is possible for discrete RL algorithms to address continuous action spaces, by regarding the multi-dimensional continuous action space as fully cooperative multi-agent reinforcement learning between dimensions, that is, each action dimension as an independent agent. Several attempts have been made by splitting a multi-dimensional continuous policy into single-dimensional discrete policies. For example, Metz et al. (2017) proposed a next-step prediction model to learn each dimension's discrete Q value sequentially; Tang & Agrawal (2020) integrated single-dimensional discrete policies with ordinal parameterization to encode the natural ordering between discrete actions; see also Jaśkowski et al. (2018) and Andrychowicz et al. (2020). It is worth remarking that these methods either cannot exploit experience replay or cannot deal with high-dimensional continuous action spaces, rendering them typically less effective than actor-critic methods for continuous control.

Our proposed D2PC structure, in some way, can be seen as a variant of multi-agent actor-critic, which relies on a centralized critic network to optimize each action dimension's discrete policy. Since the seminal contribution of deterministic policy gradients Silver et al. (2014) in 2014, off-policy actor-critic methods have arguably become the most effective way in dealing with continuous control tasks, thanks to their appropriate structure to deal with continuous actions and higher data efficiency compared with on-policy algorithms, e.g., proximal policy optimization (PPO, Schulman et al. (2015)), trust-region policy optimization Schulman et al. (2015); Wu et al. (2017), among many others. Off-policy actor-critic methods can be grouped into two categories: deterministic algorithms including e.g., the celebrated DDPG Lillicrap et al. (2015), D4PG Barth-Maron et al. (2018), TD3, and stochastic algorithms such as SQL Haarnoja et al. (2017), and SAC Haarnoja et al. (2018a). DDPG uses the Bellman equation Bellman (1966) with the temporal difference method to iteratively update the critic network, in which the loss function is given by

$$J^Q(\theta_Q) = \Big[Q(s(t), a(t); \theta_Q) - \big(r_t + \gamma Q(s(t+1), \mu(s(t+1); \theta_{\mu'}); \theta_Q')\big)\Big]^2 \qquad (1)$$

where $s(t)$ represents the state, , $a(t)$ the action, $r_t$ the reward, and $\gamma$ is the discounting factor; $\theta_Q$ and $\theta_Q'$ denote the parameters of the critic network and the target critic network; $\theta_\mu$ and $\theta_{\mu'}$ denote the parameters of the current and the target deterministic policy networks. The critic network in DDPG provides evaluations of the actions to optimize the policy of the actor network by minimizing the following loss function

$$J^\mu(\theta_\mu) = -Q(s(t), \mu(s(t); \theta_\mu); \theta_Q) \qquad (2)$$

For SAC with a stochastic policy, they developed soft RL, which introduces entropy into value and policy iterations. SAC's critic network is trained using optimizing the following loss function

$$J^Q(\theta_Q) = \Big[Q(s(t), a(t); \theta_Q) - \big(r_t + \gamma Q(s(t+1), a'; \theta_{Q'}) + \alpha \mathcal{H}\big)\Big]^2 \qquad (3)$$

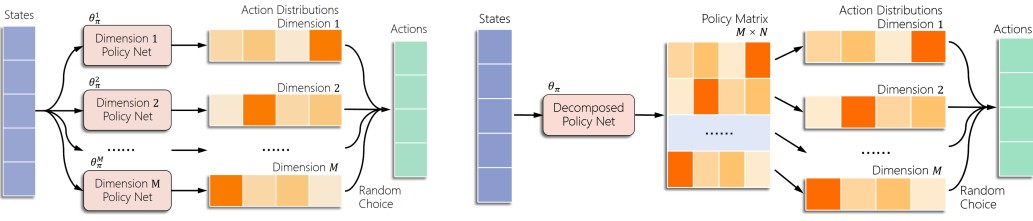

(a) Distributed discrete policy network        (b) Decomposed discrete policy network

Figure 1: Network comparison between distributed and decomposed discrete policy networks.

where the entropy $\mathcal{H} := -\log \pi(a'|s(t+1); \theta_\pi)$ with $a' \sim \pi(a'|s(t+1); \theta_\pi)$ is sampled from the stochastic policy $\pi$, and $\alpha$ is a 'temperature hyperparameter' which controls the exploration rate of the algorithm and can be automatically turned Haarnoja et al. (2018b). SAC's policy loss function is given by

$$J^\pi(\theta_\pi) = -\big(\alpha\mathcal{H} + Q(s(t), a; \theta_Q)\big) \tag{4}$$

where $\mathcal{H} = -\log \pi(a|s(t); \theta_\pi)$ with $a \sim \pi(a|s(t); \theta_\pi)$. The loss function of our SD2PC algorithm is based on SAC, which generalized its value and policy iterations to decomposed, fully cooperative multi-agent settings.

Although our D2PC structure is inspired by the framework of multi-agent actor-critic RL, we make considerable simplifications to reduce the computational complexity. In previous works like MADDPG Lowe et al. (2017), they set an actor and a critic for each agent, or COMA Foerster et al. (2018) which uses recurrent neural networks such as long-short term memory (LSTM) Hochreiter & Schmidhuber (1997) or gated recurrent units (GRUs) Chung et al. (2014) to encode the partially observed states. Our framework can be viewed as dealing with observable states and fully cooperative agents. Therefore, we can capitalize on a single critic for all action dimensions instead of one for each and every action dimension. Imitating the multi-agent actor-critic RL, we allocate a policy network (actor) for each action dimension like the structure in Figure (1a). This structure, however, requires to optimize $M$ sets of policy network parameters for a task, which still incurs relatively high computational complexity. To further reduce the complexity, we advocate a centralized decomposed discrete policy network shown in Figure (1b), which can output all the action-dimensional agents' policies simultaneously, markedly improving the algorithm's efficiency.

## 3 DECOMPOSED DISCRETE POLICY-CRITIC STRUCTURE

In this work, we focus on addressing continuous control tasks by discrete actions, where a continuous action space $\mathcal{A} := \{a \in \mathbb{R}^M\}$ with $M$ dimensions is discretized into countable discrete actions. Without loss of generality, we assume that each continuous action is restricted as $a_m \in [-1, 1]$, so that one can discretize each continuous action dimension $m$ into $N$ fractions. The resulting discrete action space $\mathcal{A}_m^d$ for the $m$-th dimension is denoted by

$$\mathcal{A}_m^d := \big\{a_{m,1}^d, a_{m,2}^d, ..., a_{m,N}^d\big\}, \quad \text{for} \quad m \in [M] := \{1, 2, \ldots, M\} \tag{5}$$

where $a_{m,n}^d$ represents the $n$-th discrete action corresponding to the action dimension $m$. The discretization above operates on a single action dimension. The discretized action space for $\mathcal{A}$ is obtained by combining all possible $M^N$ combinations of the discrete actions, namely, $\mathcal{A}^d := \{a_{m,n}^d\}_{m \in [m], n \in [N]}$ where $[N] := \{1, 2, \ldots, N\}$. Instead of setting up an output neuron for each element in $\mathcal{A}^d$, we construct for each dimension $m$ an independent discrete policy covering all the discrete actions in $\mathcal{A}_m^d$.

In previous works, there are mainly two types of methods which are compatible with experience replay, that is, DQN and actor-critic methods. We first consider using DQN to train a deterministic policy for each action dimension, given its simple structure and low computational overhead. Nevertheless, we found it is problematic to utilize DQN for each action dimension while using experience replay. The reason is as follows. Consider the Markov decision process (MDP) over the discretized action space $\mathcal{A}^d$, described as $\{\mathcal{S}, \mathcal{A}^d, p, r\}$ in which $S$ is the state space, $p$ is the transition probability function, and $r$ is the reward function. In RL algorithms using TD(0) and experience replay, the experiences

$\{s(t), a(t), r_t, s(t+1)\}$ obey the transition probability distribution function $p(s(t+1)|s(t), a(t))$, so they can be judiciously exploited to train the state-action value function. When updating each action dimension's policy independently, the MDP will degenerate into a single-action MDP $\{\mathcal{S}, \mathcal{A}_m^d, p_m, r\}$ with action space $\mathcal{A}_m^d$. As for the remaining dimensions, their polices can be viewed as fixed since independent updates are assumed across dimensions. Hence, the remaining policies can be accounted for by defining a marginal transition probability function as follows

$$p_m(s(t+1)|s(t), a_m(t)) := \mathbb{E}_{\bar{a}_m(t)}\big[p(s(t+1)|s(t), \bar{a}_m(t), a_m(t))\big], \quad \text{where} \qquad (6)$$

$$\bar{a}_m(t) = [a_1(t), \dots, a_{m-1}(t), a_{m+1}(t), \dots, a_M(t)] \text{ with } a_i(t) \sim \pi_i(a_i(t)|s(t)).$$

In order to accommodate the technique of experience replay in our single-dimensional action's update, the experiences which used to replay must obey the transition probability function $p_m$. Nonetheless, the actions $a_m(t)$ of experiences in the replay buffer are not sampled by the current policy $\pi_m(a_i(t)|s(t))$, but by some $\pi_m^{old}(a_m(t)|s(t))$, so the experiences sampled in the past do not obey the transition probability function $p_m$. That is, one cannot readily use TD(0) along with experience replay to train the per-action-dimension policy.

Concerning the issues above, we turn to the actor-critic structure with experience replay, by means of adopting a centralized critic network $Q(s(t), a(t); \theta_Q)$ to evaluate all the actions and distributed policy networks $\{\pi_m(a_m(t)|s(t); \theta_{\pi_m})\}_{m=1}^M$ instead of one-dimensional value functions for each action dimension. The critic network's value function can be updated by using experience replay, as it provides critic values for all $M$ action dimensions and its updating process is based on the MDP instead of independent single-action MDPs. As to the policy networks associated with action dimensions, they can be updated by using policy gradients with critic values provided by the centralized critic network. Each action dimension's policy loss can be given by

$$J(\theta_{\pi_m}) = \mathbb{E}_{\pi_m, \bar{\pi}_m}\big[\pi_m(a_m(t)|s(t); \theta_{\pi_m}) Q(s(t), a_m(t), \bar{a}_m(t); \theta_Q)\big] \qquad (7)$$

where, with slight abuse of notation, we write $a(t) = \{a_m(t)\}_{m=1}^M = \{a_m(t), \bar{a}_m(t)\}$ yielding $Q(s(t), a_m(t), \bar{a}_m(t); \theta_Q) = Q(s(t), a(t); \theta_Q)$ for any $m \in [M]$. Overall, corresponding to equation 7, the considered network structure with centralized critic and distributed policies is described in Figure 1(a). However, this structure has a limitation, which has $M$ groups of policy network parameters $\{\theta_\pi^1, ..., \theta_\pi^M\}$, leading to high computational complexity. In order to improve the training efficiency, we design a decomposed global policy network $\pi = \{\pi_i\}_{m \in [M]} : \mathcal{S} \to \mathbb{R}^{M \times N}$ parameterized by $\theta_\pi$. Rather than outputting each action dimension's policy by multiple individual networks, the decomposed policy network outputs a policy matrix. Each row of the matrix stands for a discretized action dimension's policy distribution. The decomposed discrete policy network's structure is shown in Figure 1(b).

The structure of the decomposed discrete policy network allows us to update its parameters simply by the stochastic policy gradient method, which samples an action $\tilde{a}(t)$ and updates the policy of all the $M$ action dimensions simultaneously as follows

$$\tilde{a}_m(t) \sim \pi_m(\tilde{a}_m(t)|s(t); \theta_\pi), \quad \text{where} \quad \tilde{a}(t) = \{\tilde{a}_1(t), ..., \tilde{a}_M(t)\}, \qquad (8)$$

$$\theta_\pi \longleftarrow \theta_\pi + \lambda Q(s(t), \tilde{a}(t); \theta_\pi) \nabla_{\theta_\pi}\Big[\prod_{m=1}^M \pi_m(\tilde{a}_m(t)|s(t); \theta_\pi)\Big] \qquad (9)$$

where $\lambda > 0$ is the learning rate.

## 4 STOCHASTIC SOFT DECOMPOSED DISCRETE POLICY-CRITIC

Due the high variance of stochastically sampled critic values and the lack of exploration strategies, updating the discrete policy with the stochastic policy gradients in equation 9 may not be effective. To improve the training efficiency of D2PC, we consider introducing soft RL and a dimension-wise value expectation technique into D2PC, which lead to our effective SD2PC algorithm. In the following, we begin by using a soft Q function with entropy $\mathcal{H} := -\sum_{m=1}^M \log \pi_m(\tilde{a}_m(t+1)|s(t+1); \theta_\pi')$ generated by the decomposed discrete policy, to update the D2PC's critic network, leading to a value loss function similar to that of the SAC algorithm in equation 3

$$J^Q(\theta_Q) = \Big[Q(s(t), a(t); \theta_Q) - (r_t + \gamma Q(s(t+1), \tilde{a}(t+1); \theta_Q') + \alpha\mathcal{H})\Big]^2 \qquad (10)$$

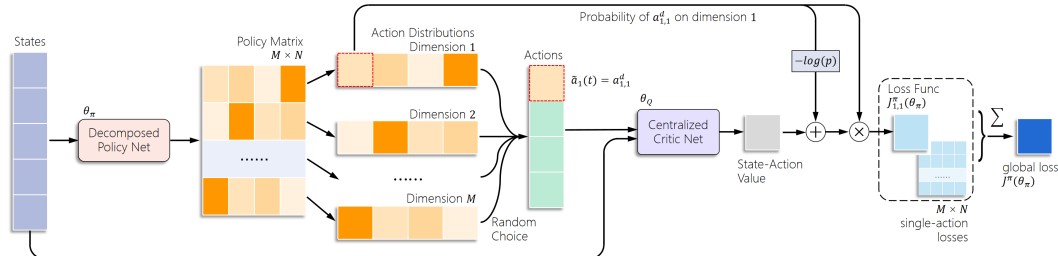

Figure 2: The computation flow of evaluating the loss function $J_{1,1}^{\pi}(\theta_\pi)$ in SD2PC, while all $J_{m,n}^{\pi}(\theta_\pi)$'s can be evaluated simultaneously in a minibatch to improve computational efficiency.

where $\widetilde{a}(t+1) = \{\widetilde{a}_1(t+1), \ldots, \widetilde{a}_M(t+1)\}$ is a global action sampled by the target policies $\{\widetilde{a}_m(t+1) \sim \pi_m(\widetilde{a}_m(t+1)|s(t); \theta_\pi')\}_{m \in [M]}$ sharing a single target policy network $\theta_\pi'$. As far as the policy update, we employ the loss function proposed in Christodoulou (2019) for discrete SAC methods based on policy gradient updates

$$J^\pi(\theta_\pi) = \mathbb{E}_{\widetilde{a}(t) \sim \pi} \left[ \alpha \sum_{m=1}^{M} \log \pi_m(\widetilde{a}_m(t)|s(t); \theta_\pi) - Q(s(t), \widetilde{a}(t); \theta_Q) \right]. \tag{11}$$

In practice however, it is impossible to evaluate the expectation involving high-dimensional random actions in equation 11, while requires us to cover the entire global discrete action space $\mathcal{A}^d$. On the other hand, if we simply draw one action $\widetilde{a}(t)$ and rely on the instantaneous loss to approximate the expected one $J^\pi(\theta_\pi)$, the resulting stochastic policy gradients will certainly have high variance. Nonetheless, we observe that the expectation of the loss function per dimension $m$ can be efficiently evaluated in closed-form. Therefore, to mitigate the high variance of stochastic sampling as well as the high computational complexity of covering $\mathcal{A}^d$, we compute a surrogate of equation 11 by changing the order of the expectation and summation, resulting a sum of expectations across dimensions, for which one only need to cover each $\mathcal{A}_m^d$ efficiently

$$J^\pi(\theta_\pi) = \frac{1}{M} \sum_{m=1}^{M} \mathbb{E}_{\widetilde{a}_m(t) \sim \pi_m} \left[ \alpha \log \pi_m(\widetilde{a}_m(t)|s(t); \theta_\pi) - (1/M)Q(s(t), \{\widetilde{a}_m(t), \widetilde{\overline{a}}_m(t)\}; \theta_Q) \right]. \tag{12}$$

The key idea behind equation 12 is to regard each action dimension as an independent agent, in that way we can first sample a global action $\widetilde{a}(t)$, then cover each and every action dimension's action space $\mathcal{A}_m^d$ to obtain the averaged sample loss. Meanwhile, the remaining dimensions' actions collectively denoted by $\widetilde{\overline{a}}_m(t)$ are viewed deterministic and replaced with their sampled values when generating the Q value $Q(s(t), \{\widetilde{a}_m(t), \widetilde{\overline{a}}_m(t)\}; \theta_Q)$ for $\widetilde{a}_m(t)$. Further, equation 12 can be equivalently rewritten as follows

$$J^\pi(\theta_\pi) = \frac{1}{M} \sum_{m=1}^{M} \sum_{n=1}^{N} J_{m,n}^{\pi}(\theta_\pi) \tag{13}$$

$$J_{m,n}^{\pi}(\theta_\pi) = \pi_m(a_{m,n}^d|s(t); \theta_\pi) \left[ \alpha \log \pi_m(a_{m,n}^d|s(t); \theta_\pi) - (1/M)Q(s(t), \{a_{m,n}^d, \widetilde{\overline{a}}_m(t)\}; \theta_Q) \right].$$

The evaluation of each $J_{m,n}^{\pi}(\theta_\pi)$ is illustrated in Figure 2, where $m = n = 1$. For each $J_{m,n}^{\pi}(\theta_\pi)$, one needs to propagate over the critic network once. Fortunately, the loss evaluations can be performed in parallel across both dimensions $m$ and discrete actions $n$ per dimension, thus accelerating the training process. In addition, the discrete policy of SD2PC can also be updated by minimizing a policy loss based on KL divergence, which is different from equation 13 but shows the same effect. We presents the details of our KL divergence policy loss of SD2PC in Appendix A.5, which can be an optional substitute of equation 13.

In terms of the exploration strategy for SD2PC, we implement the automatic temperature adjustment strategy, which was presented in the second version of SAC in Haarnoja et al. (2018b). With a prescribed target entropy $\overline{\mathcal{H}}$, SD2PC can automatically adjust the temperature hyperparameter $\alpha$ to control the algorithm's exploration rate. When $\mathcal{H} > \overline{\mathcal{H}}$, i.e., the exploration rate is excessive, SD2PC

will decrease $\alpha$, and increase $\alpha$ otherwise, by means of minimizing the following temperature loss function using gradient descent

$$J(\alpha) = \frac{1}{M} \sum_{m=1}^{M} \mathbb{E}_{\widetilde{a}_m(t) \sim \pi_m} \left[ -\alpha \log \pi_m(\widetilde{a}_m(t)|s(t); \theta_\pi) - \alpha \overline{\mathcal{H}} \right]. \tag{14}$$

Similarly to equation 12, the expectation can be approximately evaluated using

$$J(\alpha) = \frac{1}{M} \sum_{m=1}^{M} \left[ -\alpha \overline{\mathcal{H}} - \sum_{n=1}^{N} \alpha \pi_i(a_{m,n}^d | s(t); \theta_\pi) \log \pi_i(a_{m,n}^d | s(t); \theta_\pi) \right]. \tag{15}$$

We would like to remark that the target entropy $\overline{\mathcal{H}}$ in SD2PC is set for each action dimension's policy, while SAC sets a target entropy for the global policy. That is, one can interpret that Sd2PC as a scale-free hyperparameter $\overline{\mathcal{H}}$, which does not vary with the number of action dimensions $M$ in changing tasks. Nevertheless, in discrete action settings, the proper target entropy $\overline{\mathcal{H}}$ is relevant to the quantity $N$ of discretized actions, which renders $\overline{\mathcal{H}}$ hard to adjust. To address this issue, we propose an action probability transfer trick, by transferring the discrete probabilities into probability densities. With this trick, in all the baseline environments considered in Section 7, we set $\overline{\mathcal{H}} = -1.5$. More details about the trick are provided in Appendix E.2.

## 5 DETERMINISTIC DECOMPOSED DISCRETE POLICY-CRITIC

Although the proposed SD2PC method in Section 4 can tackle continuous control tasks via discrete stochastic policies, it is problematic to address continuous action space via discrete deterministic policies. When dealing with discrete deterministic settings, nether can one update the policy by optimizing the policy distribution nor adjust the action by deterministic policy gradients like DDPG Lillicrap et al. (2015), because the policy distribution is deterministic, and the discrete actions are fixed during discretization.

In order to address continuous RL via discrete deterministic policies, we further put forward a decomposed deep Q network to replace the decomposed stochastic policy network in D2PC. Instead of outputting a policy distribution matrix, the decomposed deep Q network outputs a Q value matrix, in which each row of the matrix stands for the discrete Q values $Q_d$ of all the discrete actions associated with one action dimension. Let us use $Q_d$ to denote the state-action value of $s(t)$ and discrete action $a_{m,n}^d$, while the other dimensions' actions are generated by the current policy

$$Q_d(s(t), a_{m,n}^d) := \mathbb{E}_{\mu_m, \overline{\mu}_m} \left[ \sum_{k=0}^{\infty} \gamma^k r_t \right], \quad \text{where} \quad \begin{aligned} \widetilde{a}(t+k) &= \{a_m(t+k), \widetilde{\overline{a}}_m(t+k)\} \\ \widetilde{a}_m(t) &= a_{m,n}^d \\ \widetilde{a}_m(t+k) &= \mu_m(s(t+k)) \text{ for } k > 0 \\ \widetilde{\overline{a}}_m(t+k) &= \overline{\mu}_m(s(t+k)) \text{ for } k \geq 0 \\ s(t+k) &\sim p(\,\cdot\,|s(t+k-1), \widetilde{a}(t+k-1)) \end{aligned} \tag{16}$$

in which $\mu_m$ represents the $m$-th action dimension's greedy policy, which relies on the Q value matrix generated by the decomposed deep Q network, i.e., by selecting the action having the maximum $Q_d$ value in the $m$-row of Q value matrix as follows

$$a_m(t) = \mu_m(s(t)) := \arg \max_{a_{m,n}^d \in \mathcal{A}_m^d} Q_d(s(t), a_{m,n}^d). \tag{17}$$

Let us write $\mu(s(t)) = \{\mu_1(s(t)), \dots, \mu_M(s(t))\} = \{\mu_m(s(t)), \overline{\mu}_m(s(t))\}$ to be the global policy. If the policy $\overline{\mu}_m(s(t+k))$ is fixed, then $Q_d(s(t), a_{m,n}^d) = Q(s(t), \{a_{m,n}^d, \widetilde{\overline{a}}_m(t)\})$, where $\widetilde{\overline{a}}_m(t) = \overline{\mu}_m(s(t+k))$. So, for a decomposed deep Q network with parameters $\theta_d$, we can update its deterministic greedy policy through minimizing the difference between each discrete action's Q value in the matrix and the corresponding Q value by the critic network

$$J_{m,n}^d(\theta_d) = \left( Q_d(s(t), a_{m,n}^d; \theta_d) - Q(s(t), \{a_{m,n}^d, \overline{\mu}_m(s(t); \theta_d)\}; \theta_Q) \right)^2. \tag{18}$$

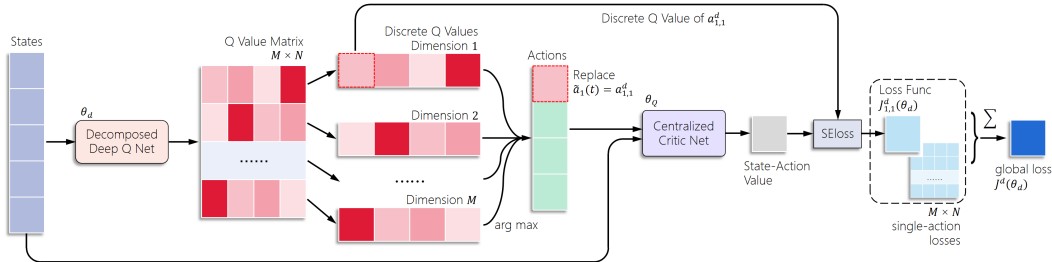

Figure 3: The computation flow of evaluating the partial loss function $J_{1,1}^d(\theta_d)$ in D3PC. Similarly, all $J_{m,n}^d(\theta_d)$'s can be obtained in parallel using mini-batches.

The flow for computing the partial loss $J_{m,n}^d$ in equation 18 is demonstrated in Figure 10 for $m = n = 1$. Similarly to SD2PC, for every action in the discrete action space, we can compute a local loss function that contributes to the global loss as follows

$$J^d(\theta_d) = \sum_{m=1}^{M} \sum_{n=1}^{N} J_{m,n}^d(\theta_d). \tag{19}$$

The update of the critic network can be done using TD(0) learning, which is similar to equation 3 in DDPG, but the target actions are generated by our proposed decomposed discrete policy. We also deploy a target policy network parameterized by $\theta_d'$ to stabilize the target policy. The loss function for D3PC's critic can is given as follows

$$J^Q(\theta_Q) = \left[ Q(s(t), a(t); \theta_Q) - \left( r_t + \gamma Q(s(t+1), \mu(s(t+1); \theta_d'); \theta_Q') \right) \right]^2. \tag{20}$$

For D3PC, we design a novel $\epsilon$-Gaussian exploration strategy $\pi_{\epsilon G}(a(t)|s(t); \theta_d)$ to promote exploration over the action space. Our $\epsilon$-Gaussian strategy first relies on $\epsilon$-greedy to select a discrete action and subsequently maps it back to the continuous action space, which, combined with a Gaussian exploration noise $\mathcal{N}(0, \sigma^2)$, is used to interact with the environment. We compared different exploration strategies in Appendix E.3, which show that our $\epsilon$-Gaussian outperforms both the $\epsilon$-greedy and the plain-vanilla Gaussian exploration strategies. In D3PC, we also employ soft target updates and TD3's double critic network Fujimoto et al. (2018). The pseudo-code of the D3PC algorithm is provided in Appendix B.

## 6 ACTOR-CRITIC WITH DISCRETE-CONTINUOUS HYBRID POLICY

In our experiments, we found that D3PC outperforms in training efficiency, but may sacrifice the final performance when comparing with continuous actor-critic algorithms, such as TD3, in baseline environments like Humanoid-v2. This observation has motivated us to combine the best of the two worlds, namely the training efficiency of discrete RL and the outstanding final performance of continuous RL algorithms. To this aim, we realize that D3PC and DDPG are only different in their policy networks, with their critic networks being the same. It is thus natural to question whether we can use the D3PC's critic network to train a continuous actor, or use the DDPG's critic network to train a decomposed discrete Q network?

Concerning the fact that we can update an actor if there is a well-trained critic network, we propose two conceptual algorithms as a manner to explore the common behind D3PC and DDPG. The first algorithm, which we call **1.D3PC-CA**, trains an additional continuous actor network along with D3PC using the same critic network, but the new actor does not contribute to the D3PC's value iterations; and, the second algorithm is **2.TD3-DQ**, that adds a decomposed discrete Q network in TD3 (a variant of DDPG), and trains the Q network using TD3's critic free from TD3's value iterations. Further details can be found in Appendix C. Experimental results of the two algorithms several benchmark tasks are depicted in Figure 4. Obviously, D3PC-CA's continuous policy performs much better than TD3-DQ in terms of final reward performances.

According to the conclusions above, we consider improving D3PC's performance by intertwining with a continuous policy. Practically, we can start by using D3PC at the beginning of the training

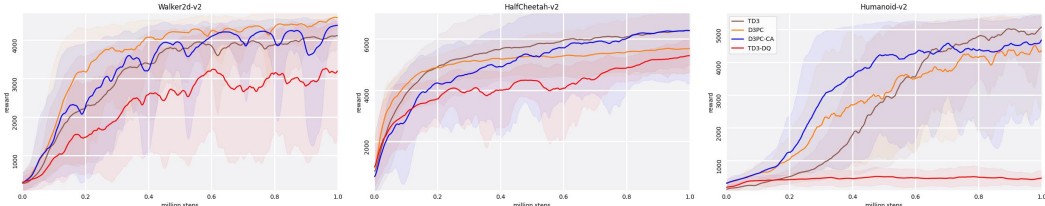

Figure 4: Total averaged rewards of DCQ-AC and TD3-CQ over 5 seeds.

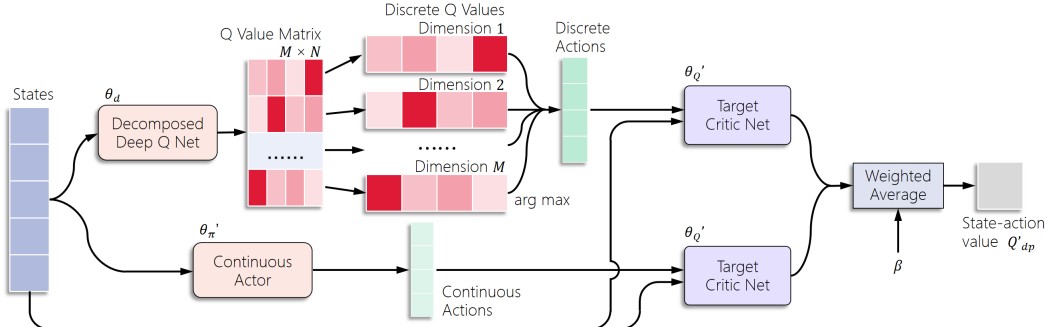

Figure 5: Critic's updates in QPC, whose target Q value is the weighted average of the Q values generated by the decomposed Q network and by the policy network.

process, then gradually shifting to an algorithm hybridizing D3PC's and TD3's policies, which is expected to inherit the initial training efficiency of D3PC as well as the final performance of TD3. Following this idea, we develop the Q-policy-critic (QPC) algorithm, which is composed of three networks, with **Q** representing a decomposed deep Q network $\mu(s(t); \theta_d)$, **P** representing a continuous policy network $\mu^p(s(t); \theta_p)$, and **C** representing a critic network $Q(s(t), a(t); \theta_Q)$. While training QPC, the continuous policy $\theta_p$ is updated using deterministic policy gradients in equation 2, and the decomposed Q network updated by equation 19. As for the critic's update, it performs TD(0) iterations with a TD target obtained by averaging two target Q values: $Q'_d = Q(s(t+1), \mu(s(t+1), \theta'_d); \theta'_Q)$ with target action generated by the Q network's policy, and $Q'_p = Q(s(t+1), \mu^p(s(t+1), \theta'_p); \theta'_Q)$ whose target action is generated by the continuous target actor. The way of computing the TD target in QPC is summarized in Figure 5, with the following value loss function

$$Q'_{dp} = \beta Q'_v + (1 - \beta)Q'_p \tag{21a}$$

$$J^Q(\theta_Q) = \left[ Q(s(t), a(t); \theta_Q) - \left( r_t + \gamma Q'_{dp} \right) \right]^2 \tag{21b}$$

where $\beta \in [0, 1]$ is a hyperparameter balancing between the continuous RL and the discrete RL. Typically, one can start the training by setting $\beta = 1$ for the first $T_\beta = 1 \times 10^5$ steps to fully exploit the training efficiency of D3PC. Afterwards, a decay rate of $\beta_- = 1 - 5 \times 10^{-6}$ is used, until a minimum of $\beta_{min} = 0.5$ is reached and kept fixed. Additionally, QPC uses a discrete-continuous hybrid policy to interact with the environment, which is found by averaging the discrete action and the continuous action as follows

$$a(t) = \beta \mu(s(t), \theta_d) + (1 - \beta)\mu^p(s(t), \theta_p). \tag{22}$$

## 7 EXPERIMENTS

The objective of our experimental evaluation is to understand how effective and efficient our proposed discrete RL methods can handle challenging continuous control tasks from the OpenAI gym Brockman et al. (2016) benchmark suite based on MuJoCo Todorov et al. (2012) environments, including Hopper-v2, Walker2d-v2, Ant-v2, Halfcheetah-v2, Humanoid-v2, and InvertedDoublePendulum-v2. Details of the baseline environments are provided in Appendix D.

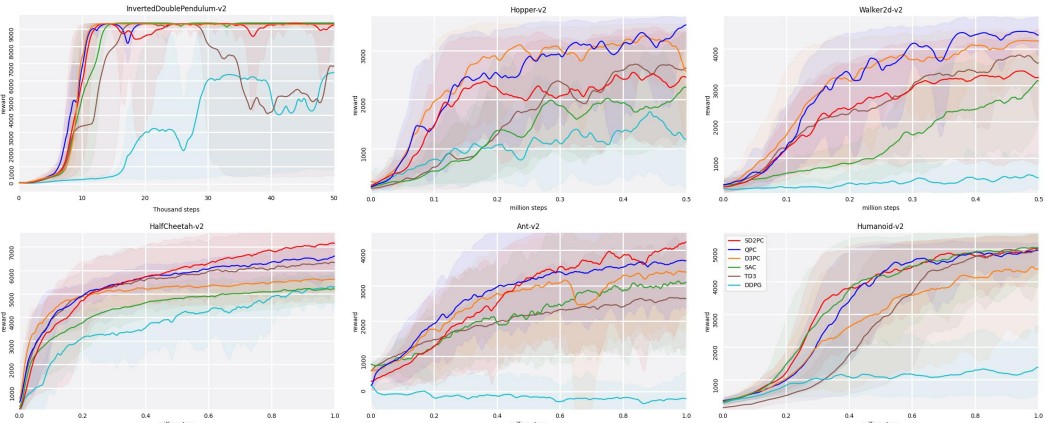

Figure 6: Training curves for the proposed algorithms, and the baselines on benchmark tasks, averaged over 5 random seeds.

We compared our SD2PC, D3PC, and QPC algorithm to the most commonly used off-policy actor-critic algorithms, including SAC Haarnoja et al. (2018b), TD3 Fujimoto et al. (2018), and DDPG Lillicrap et al. (2015). For the three baseline algorithms, we used their author-provided implementations with the original hyperparameters, except for TD3 with Gaussian exploration noise $\sigma = 0.05$ in Humanoid-v2 to avoid training failures. All of the tasks have continuous actions clipped to $[-1, 1]$, while we discretize each action dimension into $N = 20$ actions. We set the target entropy hyperparameter $\overline{\mathcal{H}} = -1.5$ for SD2PC, and $\epsilon = \sigma = 0.05$ for the exploration strategies in D3PC and QPC. Hyperparameters for the algorithms and additional network details are listed in Appendices A, B, and C. All the algorithms are realized in PyTorch Paszke et al. (2019), and evaluated on PC with one NVIDIA GPU. For each task in our experiments, we train the algorithms with Adam Kingma & Ba (2014) as optimizers, and report experimental results averaged over 5 random seeds. During the training process, every 5,000 steps (except for 500 for InvertedDoublePendulum-v2), we evaluate performances of the algorithms over five episodes by turning off the exploration strategy.

The experimental results are shown in Figure 6, in which the total average reward curves are smoothed for visual clarity. Surprisingly, the proposed methods using discrete policies exhibit similar and oftentimes competitive training performance relative to the baseline continuous RL methods in all the benchmark environments. In contrast, as observed in Tavakoli et al. (2018), existing discrete RL methods cannot even compete with the DDPG method Lillicrap et al. (2015). To the best of our knowledge, for the first time, our discrete RL methods can achieve improved performance over continuous RL methods in such challenging tasks. However, D3PC may have drawbacks in final performance, especially when dealing with complicated tasks such as Ant-v2 and Humanoid. As a remedy for D3PC, QPC fixes the drawback by mixing the discrete and continuous methods. Additionally, we made some ablation studies on the exploration policies and hyperparameters for our algorithms. Due to space limitations, these results are deferred to Appendix D.

## 8 CONCLUSIONS

This paper has considered addressing high-dimensional continuous RL tasks using discrete policies. Toward this objective, we put forth a decomposed discrete policy-critic framework, building on which three algorithms are developed: SD2PC for discrete stochastic policies, D3PC for deterministic discrete policies, and QPC for discrete-continuous hybrid policies. Substantial experimental results on benchmark tasks corroborate the efficiency of the proposed discrete RL algorithms relative to competing continuous RL alternatives including SAC, TD3, and DDPG. Despite their promising performance, we point out several directions in which the proposed methods could be improved. For example, generating the discrete action's Q values through linear interpolation instead of the critic network can reduce the computational complexity, and incorporating strategies such as including the prioritized experience replay Schaul et al. (2015), distributional value functions Bellemare et al. (2017), and multi-step temporal-difference could be effective.

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
