# OpenReview forum: "Addressing High-dimensional Continuous Action Space via Decomposed Discrete Policy-Critic"
_ICLR.cc/2023/Conference — Submitted to ICLR 2023_

### Official Review · Reviewer_AHHq · 2022-10-15

**Confidence:** 3
**Correctness:** 3
**Technical Novelty And Significance:** 3
**Empirical Novelty And Significance:** 3
**Recommendation:** 5

**Clarity, Quality, Novelty And Reproducibility:**

The paper is quite self-contained, and the authors explain well the relation between their algorithms and prior work. The Python code for the three algorithms is provided in the supplementary materials, so the results should be reproducible, in principle. (Including a Python package requirement file would help in this regard.) The manner of citation of prior work is not standard, and can be confusing - consider using parenthesis around cited references.

**Strength And Weaknesses:**

The main strength of the paper is the computational effectiveness of the proposed algorithms on well-known benchmark problems. One weakness is the need to quantize the actions, possibly leading to suboptimality of the learned policy. The third algorithm, QPC, addresses this limitation, but the empirical results show that it is not necessarily the one that always achieves the highest asymptotic reward.

Minor typos and inaccuracies:
P.1: The terms "limited actions" and "infinite actions" are not precisely defined, did the authors mean discrete and continuous actions?
P.2: "algorithms with D2PC trains fast" -> "algorithms with D2PC train fast"
P.3: "automatically turned" -> "automatically tuned"?


**Summary Of The Paper:**

The paper proposes three algorithms for finding optimal policies in environment with high-dimensional continuous controls based on decomposing the policy into independent, but cooperating policies for each action dimension, discretizing uniformly each action, and finding its optimal policy efficiently. Empirical evaluation on popular benchmarks in OpenAI Gym show faster reward improvement than well established baseline algorithms for continuous-action RL, such as SAC, DDPG, and TD3.

**Summary Of The Review:**

It appears that the three proposed algorithms have computational advantages with respect to established methods for continuous RL, such as SAC, DDPG, and TD3, so it is worth including them in the toolbox of RL practitioners. I could not find any deep insights about why these advantages are observed - intuitively, optimizing each action dimension independently could be quite suboptimal, and I wonder why this is not hampering these algorithms more. The presentation and citation style can be improved a bit.

---

> ### Comment · Reviewer_AHHq · 2022-11-30
> **Score revised**
>
> I see that nobody else among the reviewers recommended accepting the papers, and would not disagree with rejecting it. I have revised my score downwards accordingly.

---

### Official Review · Reviewer_yKnh · 2022-10-20

**Confidence:** 3
**Correctness:** 2
**Technical Novelty And Significance:** 2
**Empirical Novelty And Significance:** 2
**Recommendation:** 3

**Clarity, Quality, Novelty And Reproducibility:**

* The quality of the paper is beyond average.
* The clarity of the paper is low and the structure of the write-up can be drastically improved.
* I'm not quite sure about the originality of this work because I'm feeling like this paper tells a simple story in a very complex way. If my understanding is correct (see my summary), then the methodology itself is not quite novel but the empirical findings are interesting.
* For reproducibility, authors uploaded two core python files but they cannot be run out-of-the-box and no further instructions about how to use these two files are provided.


**Strength And Weaknesses:**

Strength
* Discretization and independent modelling allows discrete approaches to be applied continuous task without too much efforts.
* The empirical results are somewhat surprising since a potential weakness of such independent modelling is just about whether it can be scaled up to high dimensionality.

Weakness
* The experiments didn't answer any concrete research questions. The paper proposed a new way of discrete action modelling but how does this compare to previous discretization/modelling methods? What if you still model action dimensions autoregressively or jointly? Are they going to be slow? Or maybe they can even perform worse (it will be surprising if it's indeed the case)? If the main claim is your discretized methods perform better than continuous agents, the experiment should provide more insights about why that's the case because it's not intuitive.
* The presentation of this work is quite confusing. Authors claim the method is motivated by multi-agent RL, which is an unnecessary detour. As the authors mentioned, action discretization itself is not a novel idea. The problem is how we can model the joint distribution of the discretized dimensions. One advantage of discretization is we can easily model multiple modes of the action distribution while it also blows up the size of the action space. One way to avoid the blowup is to model action dimensions auto-regressively (Metz et al). This paper basically says that just assuming each dimension is independent also gives a decent performance.
* This paper didn't clearly describe the relationships with prior works. It simply says
> It is worth remarking that these methods either cannot exploit experience replay or cannot deal with high-dimensional continuous action spaces, rendering them typically less effective than actor-critic methods for continuous control

This is confusing because I'm not sure why discretization and action modelling is tied to replay buffers. Aren't these just some orthogonal design choices?

* The write-up of the methodologies reads like a technical report that simply describes the development of the whole project. For example, there are a lot of sentences like:

> We first consider using DQN to train a deterministic policy for each action dimension, given its simple structure and low computational overhead. Nevertheless, we found it is problematic to utilize DQN for each action dimension while using experience replay.

In the methodology section, readers expect a structured description of the final version of your algorithm. If other natural design choices are important for the story, it would be better to discuss them in the ablation.

* Tons of low-level details are coupled with high-level ideas throughout the whole paper which is distracting.


**Summary Of The Paper:**

This paper proposes to discretize the continuous action space and model each dimension independently. Authors claim this idea is motivated by multi-agent RL and each agent corresponds to a dimension of the high-dimensional action space. Three variations of algorithms motivated by the idea are proposed that are SD2PC, D3PC and QPC. SD2PC is based on SAC, which is an actor-critic algorithm with policies modelled explicitly. D3PC is based on DQN which only models Q values therefore the policy is deterministic if one greedily selects the action with the highest Q value. QPC mixes continuous and discrete components, which leads to better performance.
The empirical evaluations on gym-locomotion control tasks show that discretized agents can be as good as methods that are directly operating on continuous space and the mixed agent can perform slightly better than both continuous and discrete variation of agents.

**Summary Of The Review:**

I think the paper does have some potential. I guess the authors want to tell a story like Q-MIX that simple methods with seemingly strong assumptions work surprisingly well. But I believe the current quality of the paper didn't match the ICLR standard, especially the write up of the paper should be improved in future submission.

---

### Official Review · Reviewer_hBbu · 2022-10-22

**Confidence:** 5
**Correctness:** 3
**Technical Novelty And Significance:** 2
**Empirical Novelty And Significance:** 2
**Recommendation:** 3

**Clarity, Quality, Novelty And Reproducibility:**

I have concerns over novelty of ideas. I believe the exposition of ideas needs improvement as well. The story of the paper is not clear as it stands, with many critical missing references. Some other discussions are not very well written, e.g. the exposition of the non-stationarity problem of dealing with Q-learning in factored action spaces (Sec 3) was quite unclear, and failed to relate to many known literature around the issue from cooperative multi-agent RL. Evaluation is also not super convincing to me. Nowadays, many higher dimensional continuous control domains are available (through DM Control Suite for instance), but this paper only evaluates in v2 of OpenAI Gym's MuJoCo domains. Evaluating on 5 random seeds, and for only 1M steps is also somewhat limited (10 seeds and 3M steps is more common for these domains). Overall, I feel the paper is not ready for publication at this instance and requires significant positioning in light of the missing and the not-fully-covered literature.

**Strength And Weaknesses:**

**Strengths:**
- The paper focuses on an interesting problem.

- The illustrations really help understand the ideas.

- I think this paper would become a valuable resource as a one-stop coverage of all existing and potentially some novel algorithmic combinations for addressing continuous control problems through discretization (assuming all missing related works get incorporated appropriately and the missing discussions get added in great details).


**Weaknesses:**
- The argument for QL/DQN with Experience Replay not being compatible with factored action-space representations (discussed in Sec 3) is well known (especially in the multiagent RL literature). Yet, Tavakoli et al. (2018) have shown that this combination can still perform competitively with DDPG, even significantly outperforming it in the high-dimensional task of Gym's Humanoid. But unfortunately, despite citing said paper, relevant discussions of it (or the details of the approach) are missing in the paper. In fact, using a similar parameter-sharing scheme in the way of Fig 1(b) was used in the approach proposed in the said paper, and also they used a centralized state-value critic (as in dueling networks) to make the training more centralized.

- How does the approach in Sec 3 differ from what was proposed in Tang & Agrawal (2020)? It's basically the result of the policy gradient theorem, used with a factored policy representation (using the independence of policies' assumption).

- In fact, a COMA-style PG update would have lower variance without any bias wrt. the method proposed in Sec 3 (see [1]).

- Important/critical related works are missing such as [2,3,4,1], and some other important related works have been included but very shallowly such as Metz et al. (2017), Tavakoli et al. (2018), Tang & Agrawal (2020).

- How does the approach of Sec 4 differ in principle from the SAC variations in Ref. [4]? They used distributions over two and three subactions per action-dimension, but the same update rule can be used to train with higher number of subactions per action dimension.

- Comparison with the Amortized Q-learning algorithm of Ref. [3] is necessary in my view. Also, a detailed discussion of the difference of said approach wrt. those presented in this paper would be important.

- Comparison to a well-tuned agent from Tavakoli et al. (2018) and HGQN (r=1) from Ref. [2] would be necessary. Approach of Sec 5 is similar to Branching DQN by Tavakoli et al. (2018), with the difference that the target is a joint-action Q-estimator. Whether such a join-action Q-estimator would work better or not needs to be empirically evaluated. To compare, you'd have to use a similar architecture to that used in your paper, spend the same strategy to tune its hyperparameters to see whether there is a clear advantage.

- Claims such as "*To the best of our knowledge, for the first time, our discrete RL methods can achieve improved performance
over continuous RL methods in such challenging tasks.*" clearly illustrate that the authors should closely read the suggested related works.


**References:**

[1] Wu et al. (2018) "Variance Reduction for Policy Gradient with Action-Dependent Factorized Baselines." ICLR.

[2] Tavakoli et al. (2021) "Learning to represent action values as a hypergraph on the action vertices." ICLR.

[3] Van de Wiele et al. (2020) "Q-Learning in enormous action spaces via amortized approximate maximization." arXiv.

[4] Seyde et al. (2021) "Is Bang-Bang Control All You Need? Solving Continuous Control with Bernoulli Policies." NeurIPS.

**Summary Of The Paper:**

This paper proposes several methods for addressing continuous control problems by using discrete-action methods, in such a manner that they scale gracefully with increasing action dimensionality.

**Summary Of The Review:**

**Minor Questions:**

- In Par 1 of Sec 2, it is mentioned that previous approaches, e.g., Tavakoli et al. (2018) or Metz et al. (2017) are not compatible with experience replay or cannot deal with high-dimensional tasks. From the experiments of both these papers, I see that they both experiment with the Humanoid task (the highest dimensional task in your paper) and they both use experience replay. Can you comment in what specific context you mean they are incompatible?

- What is the difference between Fig. 1(a) and 1(b)? From what I understand, 1(a) has no shared parameters, while 1(b) has all shared parameters except for the output linear layer, correct?


**Minor:**

- `\citet` and `\citep` are used mistakenly throughout the paper. E.g. "*Since the seminal contribution of deterministic policy gradients Silver et al. (2014) in 2014* [...]", here you need to display *(Silver et al., 2014)* by using `\citep`.

---

### Official Review · Reviewer_zjS6 · 2022-10-23

**Confidence:** 4
**Correctness:** 2
**Technical Novelty And Significance:** 2
**Empirical Novelty And Significance:** 2
**Recommendation:** 3

**Clarity, Quality, Novelty And Reproducibility:**

- I found it a bit odd that the baseline performance reported in Figure 6 is worse than the performance reported in the original paper (e.g., TD3 and SAC on HalfCheetah, Ant and Walker2d). It would be helpful if the authors could comment on the discrepancy in the performance (e.g., was it due to the gym version difference?). In addition, the authors mentioned that "the the total average reward curves are smoothed for visual clarity" but did not say how the smoothing was done.

- One of the main method QPC uses an *average* of the continuous policy and the discrete policy in the action space for exploration. I found this design decision to be quite odd as the interpolation of two good actions in the action space might not lead to actions that are good. There are some simple alternatives that could have been used here: 1) selecting continuous policy with $1-\beta$ probability and discrete action otherwise, 2) picking the one with the maximum Q value.



**Strength And Weaknesses:**

*Strength*:

The paper is generally well-written and easy to follow.

*Weaknesses*:

I found the empirical results not strong enough to support the main claims of the paper (that their RL algorithms can match and improve over continuous actor-critic methods). First of all, I have doubts over the validity of the baseline performance reported in Figure 6 (see more details below) which makes it hard to judge how much benefits that discretizing actions could actually bring. In addition, the QPC was motivated to make its final performance  better by transitioning from discrete policies to a continuous policy; however, I did not find statistical significant evidence that this is true in Figure 6 (QPC works very similar to SD2PC, D3PC on all tasks).

**Summary Of The Paper:**

The authors propose three new RL algorithms for tasks with continuous action space. The first two algorithms involve the core idea of discretizing the action space in a scalable manner where each action dimension is discretized independently and the action selection is independent as well. Such discretization allows the authors to convert the original problem with high-dimensional action space into a multi-agent RL problem with different discrete policy for each action dimension and a shared critic for all action dimensions. Although these  two algorithms are fast in the beginning of training, they can lead to worse final performance due to discretization. To address the sub-optimal final performance issue, the author propose the final algorithm (QPC) that use a combination of discrete policies and a continuous policy throughout the training. This final algorithm enjoys the benefits of RL with discrete actions while matching the final performance of prior continuous action space RL algorithms.

**Summary Of The Review:**

While I do find the proposed methods to be conceptually interesting and that it could potentially be scaled up to higher dimensional problems, they do occur to me right now as complicated methods with little to no improvements over existing methods. As the current state of the paper, I would not recommend acceptance.

It is possible that the methods are able to perform much better than baselines in higher dimensional problems (right now the environment with the largest action space is only 17-dimensional), and showing such results could greatly strengthen the paper especially around the point that the discretization proposed in the paper is more scalable.

---

### Official Review · Reviewer_zaix · 2022-10-24

**Confidence:** 4
**Correctness:** 4
**Technical Novelty And Significance:** 3
**Empirical Novelty And Significance:** 3
**Recommendation:** 3

**Clarity, Quality, Novelty And Reproducibility:**

- Clarity: average-low, see weak points above
- Quality: high, the proposed algorithms are sound, well-motivated and perform well in the experiments
- Originality: high
- Reproducibility: seems high, the formulas are given and the figures clearly show the architecture of the networks considered in the paper.

**Strength And Weaknesses:**

Strengths:

- The paper presents many novel ideas, propose algorithms that appear sound, and exhibits strong empirical evidence that the proposed algorithms work and outperform strong baselines. This is clearly a paper that teaches something to the readers and is interesting to read.

Weaknesses:

- The paper is quite difficult to read and to understand. It is possible to understand the contributions, but there are so many things in this paper (the stochastic algorithm for instance, while QPC is built on the deterministic algorithm) that understanding what is the contribution and what is the path towards it is difficult.

From my understanding, SD2PC is only used to later build D3PC, and hence is part of the explanation, not one of the contributions. I would therefore maybe move its description in an appendix, or better present it as the general idea: from a state, we compute a per-action-dimension discrete action, that is then merged into a full action and used to train the critic. Then we move to D3PC in which the per-action-dimension actions are generated by taking the argmax over Q-Values.

The same applies to QPC, that seems to be the one contribution of this paper, and whose general idea is to mix two deterministic actors: one based on discrete actions, one on continuous actions. Both actors are trained in parallel from a centralized critic, and which actor we use for training the critic (and action execution) changes over time, to privilegiate the discrete one in early stages of learning, then progressively move to the continuous one.

I think that mentioning (in the authors's word and with higher accuracy) what I explain above, in the introduction or even the abstract, would greatly help the readers understand what this paper is about, and navigate it. It would also explain from the beginning something that nows appears a bit too suddenly: the "let's now use DDPG's critic network with our actor and some other continuous actor".

**Summary Of The Paper:**

The paper proposes Reinforcement Learning algorithms for continuous actions based on per-dimension discretization and a centralized critic network. The aim of the algorithms is to allow for sample-efficient discretized RL, without the curse of dimensionality it incurs. The work is inspired from multi-agent systems, in which a centralized critic (that observes everybody's action) is used to train the separate actors of the agents.

The proposed algorithms outperform strong baselines on challenging environments.

**Summary Of The Review:**

Very interesting idea, but whose presentation is difficult to follow. The interesting idea really deserves to be published at a top conference. But the presentation really makes this paper difficult to follow (and thus not to accept). I'm currently recommending rejection, but an updated version of the paper with a clearer path towards QPC would allow me to recommend acceptance.

---

### Decision · Program_Chairs · 2023-01-20

**Decision:**

Reject

**Justification For Why Not Higher Score:**

There are many significant deficiencies pointed out by the reviewers, which the authors did not rebut.

**Justification For Why Not Lower Score:**

N/A

**Metareview: Summary, Strengths And Weaknesses:**

The work proposes a method for discretizing continuous actions without incurring the curse of dimensionality. To achieve this, the authors propose discretizing and choosing different action dimensions separately with different but shared policies but maintaining a single critic. However, there are many limitations of the work. The advantage of learning discretized actions with actor-critic methods needs to be clarified when continuous action can already be learned with actor-critic directly. Some reviewers pointed out that the paper was hard to follow, and the baseline performances did not match those reported in the original paper. Moreover, the advantage is unclear when both continuous and discrete actors are combined, as the computation and memory used will be strictly more than either alone. I recommend addressing these issues and improving the paper substantially for the next venue.